# Glycaemic Variability and Risk Factors of Pregnant Women with and without Gestational Diabetes Mellitus Measured by Continuous Glucose Monitoring

**DOI:** 10.3390/ijerph18073402

**Published:** 2021-03-25

**Authors:** Martina Gáborová, Viera Doničová, Ivana Bačová, Mária Pallayová, Martin Bona, Igor Peregrim, Soňa Grešová, Judita Štimmelová, Barbora Dzugasová, Lenka Šalamonová Blichová, Viliam Donič

**Affiliations:** 1Department of Medical Physiology, Faculty of Medicine, Pavol Jozef Safarik University, 040 11 Košice, Slovakia; ivana.bacova@upjs.sk (I.B.); maria.pallayova@upjs.sk (M.P.); martin.bona@upjs.sk (M.B.); igor.peregrim1@upjs.sk (I.P.); sona.gresova@upjs.sk (S.G.); judita.stimmelova@upjs.sk (J.Š.); viliam.donic@upjs.sk (V.D.); 2Internal and Diabetology Outpatient Department, Human-Care s.r.o., Affiliated Study Foundation for Faculty of Medicine, Pavol Jozef Safarik University, 040 11 Košice, Slovakia; vierkaster@gmail.com; 3Department of Medical and Clinical Microbiology, Faculty of Medicine, Pavol Jozef Safarik University, 040 11 Košice, Slovakia; barbora.dzugasova@student.upjs.sk; 4Department of Pathological Physiology, Faculty of Medicine, Pavol Jozef Safarik University, 040 11 Košice, Slovakia; lenka.blichova@student.upjs.sk

**Keywords:** continuous glucose monitoring, gestational diabetes mellitus, glycaemic variability, risk factors

## Abstract

*Background*: The aim of the study was to compare the continuous glucose monitoring (CGM)-determined glycaemic variability (GV) of pregnant women with gestational diabetes mellitus (GDM) and without GDM (CG; control group). The secondary aim was to evaluate the association between risk factors of diabetes in pregnancy and parameters of glyceamic control. *Methods*: Demographic, biometric and biochemical parameters were obtained for pregnant women (20–38 years old) who after an oral glucose tolerance test were examined by 7-day continuous glucose monitoring using a iPro^®^2 Professional CGM. *Results*: The differences in GV between women with GDM and CG compared by total area under glucose curve (total AUC, (mmol·day/L) was statistically significant (*p* = 0.006). Other parameters of glycaemic control such as mean glucose, standard deviation, coefficient of variation, J-index, % time-above target range 7.8 mmol/L (%TAR), % time-in range 3.5–7.8 mmol/L (%TIR), time-below target range 3.5 mmol/L (%TBR), glycated haemoglobin were not significantly different in the study groups. Risk factors (a family history of diabetes, pre-pregnancy BMI, higher weight gain and age) correlated with parameters of glycaemic control. *Conclusions*: We found a significant difference in GV of women with and without GDM by total AUC determined from CGM. TIR metrics were close to significance. Our work points at an increased GV in relation to the risk factors of GDM. Pregnant women with risk factors have higher probability of severe GV with its consequences on maternal and fetal health state.

## 1. Introduction

Continuous glucose monitoring (CGM) provides unique insights into daily glycaemic control and permits a better understanding of how glucose variability may influence acute and long-term complications of diabetes. The CGM sensor is applied to the back of a patient’s upper arm and records glucose levels every 10 s. This detailed information allows immediate and personalized therapy [1,2].

Although CGM has been used successfully in type 1 diabetes (T1D) and type 2 diabetes (T2D) patients, the effectiveness of CGM in improving pregnancy outcomes complicated by gestational diabetes mellitus (GDM) is still understudied [3].

GDM is defined as any degree of glucose intolerance with onset during pregnancy. It is associated with increased feto-maternal morbidity as well as with long-term complications in the mother and offspring [4]. 

The amount of insulin secreted by the beta cells of the pancreas in GDM was found to be lower than that of pregnant women with normal glucose tolerance. The highest risk pregnancies are characterized as pregnant women with GDM, women with the presence of polycystic ovarian syndrome, obesity, pre-existing GDM, a family history (FH) of diabetes, as well as women with glycosuria during pregnancy [5,6].

The risk of complications in women with GDM increases proportionally with the worsening of glycaemic tolerance. Comprehensive glucose monitoring and therapy are indispensable to prevent these complications because even small increases in maternal glucose are related to poorer clinical outcomes [7,8]. The most common complications among children of women with metabolic abnormalities during pregnancy include: the future risk of obesity, impaired glucose tolerance or the development of T2D [9,10]. For this reason, glycaemic targets are very strict during pregnancy. 

Women with GDM must cope with several factors affecting their pregnancy in the short period of time. These include understanding of the disease, coping with the blood glucose measurement, starting diet therapy, regular physical activity, and eventually subcutaneous insulin therapy with self-management under regular supervision. Self-monitoring of blood glucose (SMBG) is a frequently used method for measuring blood glucose during pregnancy. However, this method is not sufficient to detect all glycaemic fluctuations [11].

There is emerging interest in using CGM in pregnancy. CGM was found to allow better estimation and description of short-term glycaemic compensation in comparison with HbA1c [12]. CGM data helps to look “beyond” glycated haemoglobin (HbA1c). CGM defines short-term glycaemic compensation for an individual patient for practice but also for research [13]. The recent recommendations from the international consensus on time in range decided that the important metrics for CGM are the percentage of time spent-in –range (%TIR), time-above-target range (%TAR) and time-below-target range (%TBR) [14].

The primary objective of the present study was to compare the CGM-determined GV of pregnant women at the turn of the 2nd and 3rd trimesters with and without GDM. The secondary objective of the study was to evaluate the association between risk factors of diabetes in pregnancy and parameters of glycaemic control. 

## 2. Materials and Methods

The study was conducted in 23 pregnant women (20–38 years old) that were recruited at the turn of the 2nd and 3rd trimesters immediately following an oral glucose tolerance test (OGTT) during the study period 2019–2020. All subjects provided written informed consent before participating in the study. The study was approved by the Ethics Committee of the Košice Self-Governing Region, protocol number: 8N/2019. The research was carried out in accordance with the principles of the Declaration of Helsinki.

Included participants were diagnosed with GDM when they met criteria of the Slovak Diabetes Society 2018: 75 g 2-h OGTT: fasting glucose ≥ 5.1 mmol/L, 1-h glucose ≥ 10.0 mmol/L, 2 h glucose ≥ 7.8 mmol/L.). If they did not meet the criteria for GDM, they were denoted as pregnant women without GDM [15]. Non-pregnant women, pregnant women with T1D, T2D, other specific types of diabetes and non-compliant women were excluded from the study. 

Following the OGTT, pregnant women were invited to our research center and their demographic, biometric and biochemical parameters were obtained before examination using a 7-day CGM device (iPro^®^2 Professional, Medtronic Diabetes, Northridge, CA, USA). The CGM glucose recordings were calibrated with preprandial capillary blood glucose measurements obtained by the participant at least three times a day. These were the fasting morning glucose, the glucose before teatime, and the glucose before sleep. Participants recorded all meals and physical activity. Blood glucose was measured by a Contour^®^ Link glucose meter (Bayer, Whippany, NJ, USA). Three out of 23 participants had inaccurate CGM results due to the sensor’s low intrinsic signal, and they were excluded from analyses. Half of the women were primigravid, and the other half were multigravida. The investigated group was divided into two subgroups based on the outcomes of the OGTT: (1) GDM group and (2) group without GDM–control group (CG).The performance and accuracy of the iPro2 CGM system was assessed by using the optimal accuracy criteria of the CGM glucose data which was calculated by the CGM software of the glucose sensor and the glucose meter data [16]. 

### The Investigated CGM Parameters

The investigated CGM measures and indicators included:(1)The area under the interstitial glucose concentration curve (AUC) for the entire day (total AUC) in mmol·day/L, calculated by the expression Total AUC =$\;\left(\bar{x}\;\times \;n\;measurements\right)÷\;\left[\left(84\times \;n\;measurements\right)÷2016\right].\;$(2)The hypoglycaemic area under the interstitial glucose concentration curve below 3.5 mmol/L normalized for a 24-h period (AUC_below3.5_) and hyperglycaemic area under the interstitial glucose concentration curve above 7.8 mmol/L normalized for a 24-h period (AUC_above7.8_) in mmol·day/L.(3)The 7-day mean blood glucose levels and glucose standard deviation representing the most common CGM parameters of glucose compensation.(4)J-index, which is an alternative parameter of glucose control evaluation designed to stress the importance of the mean level of glycaemia and the GV [17]. The J-index equation is as follows:

J-index = 0.001 × (mean + SD)^2^(5)The coefficient of variation (%CV) which is defined as the ratio of the SD to the mean glucose [18] and calculated by the formula:$$\%\mathrm{CV}\;=\;\mathrm{SD}/\bar{x}\times 100$$(6)%TIR, %TAR, %TBR. The target values of %TIR, %TAR and %TBR for GDM in pregnancy as determined by the international consensus are presented in Table 1.


Demographic characteristics and outcome data were summarized with frequencies and percentages for categorical variables and with mean ± standard deviation for continuous variables. Median (interquartile range) was used for the graphical record of total AUC in compared groups. Bivariate analysis (non-parametric Spearman’s correlation) was utilized to examine relationships between CGM outcomes and their potential predictors. Findings were statistically significant at the 5% level. Statistical analyses were performed using Stata Special Edition Version 13.1 (StataCorp LP, College Station, TX, USA). 

## 3. Results

The investigated group was divided into two subgroups based on the outcomes of the OGTT-into (1) GDM group and (2) group without GDM-control group (CG). Mean ± standard deviation: (1)GDM: (GDM, *n* = 11) gestational week 29.1 ± 2.34, age 30.6 ± 4.76 years, HbA1c: 5.52 ± 0.34%, pre-pregnancy body mass index (BMI) 28.2 ± 3.13 kg/m^2^(2)CG: (CG, *n* = 9), gestational week 29.33 ± 3.0, age 29.6 ± 2.56 years, HbA1c: 5.25 ± 0.2%, pre-pregnancy BMI (kg/m^2^) 26.5 ± 3.42 kg/m^2^

The graphical expression of the OGTT in the GDM group and CG is presented in Figure 1.

The baseline characteristics of study participants (*n* = 20) are presented in Table 2.

In the GDM group, there were six primigravid and the rest five women were multigravid. In the CG, there were four primigravid, the remaining five women were multigravid. All pregnant women in CG were normotensive. In the GDM group, there were two women on antihypertensive treatment and this hypertension had first occurred in pregnancy. The CGM indicators of GV in investigated groups (GDM/CG) of pregnant women are presented in Table 3.

The difference in GV between GDM and CG compared by total AUC (mmol·day/L) was statistically significant (*p* = 0.006), Figure 2. 

Other parameters of GV such as SD, %CV, J-index, AUC_above7.8_, AUC_below3.5_, %TAR, %TIR, %TBR were not statistically different in the investigated groups. 

The mean %TIR in pregnant women with GDM was 94.63%, the mean %TIR in the CG was 98.17%. The mean %TAR in pregnant with GDM was 5.36%, but only 73% of women met the target for %TAR (<5% of time above target 7.8 mmol/L). The mean %TAR in the CG was 1.67% and all of them reached glycaemic goals. %TBR was 0 in GDM group. One woman without GDM had blood glucose value below 3.0 mmol/L.

Non-parametric Spearman’s correlation was performed to explore associations between the spectrum of risk factors and glycaemic control parameters in all participants. FH of diabetes was positive in almost 91% of women with GDM. In the CG, there were three women (33.3%) with FH of diabetes. FH of diabetes significantly correlated with Total AUC (mmol·day/L) (*p* = 0.019) as well as with fasting glycaemia from OGTT (*p* = 0.035).

The GV in pregnancy expressed by %TAR correlated with pre-pregnancy BMI (*p* = 0.054). 

Higher weight gain in pregnancy significantly correlated with the coefficient of variability (*p* = 0.030).

The age of pregnant women (women aged ≥30) correlated negatively with %TIR (*p* = 0.003), positively with %TAR (*p* = 0.003) and AUC above 7.8 (mmol·day/L) (*p* = 0.042) (Table 4).

## 4. Discussion

### 4.1. CGM/HbA1c in Pregnancy

The primary objective of the present study was to compare the CGM-determined GV of pregnant women at the turn of the 2nd and 3rd trimesters with and without GDM. Our findings demonstrate that GV represented by total AUC was significantly higher in the GDM group compared to CG. Total AUC reflects not only duration and magnitude of glycaemia but also the severity of hyperglycaemia. 

In contrast, routinely used HbA1c didn’t show any significant difference between compared groups (GDM/CG). This is consistent with a recent study which showed that in pregnancy complicated by GDM, the HbA1c had poor reliability and insufficient sensitivity or specificity for diagnosis [19]. The result is same as for the studies which reported that HbA1c values cannot replace OGTT for the diagnosis of GDM [18].

Several studies have shown a high efficiency of HbA1c for the diagnosis of GDM in which the AUC values ranged from 0.805 to 0.937 [20,21,22]. Although HbA1c reflects the average blood glucose level, it is not the most complete expression of blood glucose. For example, it does not reflect other characteristics of blood glucose control such as increasing or decreasing the risk of complications. It does not reflect the acute changes of blood glucose, the range of glucose changes during day, and it can´t reflect blood GV [23].

We also found that TIR parameters were more exact predictors of GV in pregnancy than %CV, SD or mean 7-day CGM glucose. In accordance with the findings by Vigersky et al. we suggest that %TIR, %TAR and %TBR are important parameters of CGM evaluation [24]. TIR can evaluate the immediate effect of therapeutic changes, glycaemic excursions and the time spent in a “safe” zone [12].

### 4.2. CGM/SMBG in Pregnancy

Self-monitoring of blood glucose (SMBG) is an important tool to manage diabetes during pregnancy. However, proper implementation of SMBG in pregnant women requires understanding of its applications and limitations (efficacy and accuracy). Current updated evidence suggests that CGM is superior to SMBG among GDM pregnancies in terms of detecting hypoglycaemic and hyperglycaemic episodes including nocturnal hyperglycaemia, which might result in an improvement of maternal and fetal outcomes [3,25].

### 4.3. Family History of Diabetes

In our participants with GDM, almost 91% of women had a family history (FH) of diabetes that correlated with total AUC. A FH of diabetes in the first-degree relatives is one of the major risk factors for development of GDM. This suggests the involvement of genetic predisposition in the pathophysiology of GDM. Irving et al. reported a 31% prevalence of GDM in case of a positive FH of diabetes in pregnant women [26].

### 4.4. Pre-Pregnancy BMI

In our group, GV correlated with pre-pregnancy BMI. Higher BMI before pregnancy correlated positively with %TAR. Similarly, in a Brazilian retrospective case-control study, pre-gestational obesity (*p* = 0.001), sedentarism (*p* = 0.034), and excessive maternal weight gain were in relation to constant GV [27]. Many adverse complications for mother or offspring (large for gestational age) are associated with higher pre-pregnancy BMI which is probably caused by low adiponectin level [28]. Some studies suggest that the lower plasma adiponectin concentration in early pregnancy may be associated with subsequent development of GDM [29,30]. Fortunately, pre-pregnancy BMI is one of the few influenceable factors in preconception care [31]. 

### 4.5. Maternal Weight Gain

Weight gain during pregnancy is an important risk factor for increased GV in the GDM. In our study, weight gain correlated positively with the %CV. The weight gain during pregnancy is associated with the premature labour, foetal macrosomia, as well as intrauterine growth restriction [32]. Recommendations for weight gain are dependent on pre-pregnancy BMI. In women with T1D in pregnancy, it has been proved that the weight gain above the recommended values may lead to a significant increase in risk of foetal macrosomia: 42% of large for gestational age compared to 8% in women with the recommended weight gain [33].

### 4.6. Maternal Age

We observed that older pregnant (≥30 years) women had higher GV (%TIR, %TAR, AUC above 7.8). The age of women in pregnancy is one of the best studied risk factors for development of GDM. A study in Iran and Bahrain identified women’s age as one of the major risk factors for GDM [34]. Several studies confirmed that pregnant women aged 35–40 years of age had a significantly higher risk of GDM compared to younger women (OR 2.63, 95% CI 2.4–2.89). In Canada, researchers led a study with 111.563 pregnant women. They found that 22.4% of pregnant with GDM were older than 35 years of age, compared to only 10.3% in the same age without diabetes [35].

The strength of our study was that we found the significant difference in glycaemic control of women with and without GDM by total AUC determined from CGM. Total AUC was different even when routinely used HbA1c showed almost zero difference in investigated groups. Another important outcome of the study was that GV once measured by CGM was associated with various risk factors of GDM e.g., FH, weight gain or pre-pregnancy BMI. 

However, several potential limitations of our study need to be considered. The measurement of interstitial glucose by CGM may not precisely reflect the levels of blood glucose. The iPro2 device used includes the physiological gap between the interstitial and blood glucose readings (10–15 min) and the dependence on the accuracy of the glucose meter used.

Due to the cross-sectional nature of this study, we cannot conclude on the causal nature of the relationship between GV severity and risk factors of GDM. Other limitation of this study was that we did not evaluate detailed dietary information in relation to GV in investigated groups. 

Measures of visceral adiposity and the percentage of body fat were not obtained for this study and we acknowledge that these informations could be important when speaking about risk factors in pregnancy complicated by GDM. Last, very important limitation was that the study was small, the sensor’s low intrinsic signal and/or missing calibrations in some of the participants resulted in missing or inaccurate CGM data that were excluded. 

## 5. Conclusions

To conclude, we found the significant difference in glycaemic control of women with and without GDM by total AUC determined from CGM. Total AUC was different even when the routinely used HbA1c parameter showed almost zero difference between the investigated groups. TIR metrics were close to significant difference between GDM and CG. These findings suggest the need for detailed glyceamic control measurements in pregnancy complicated by GDM. Larger studies could prove the importance of TIR metrics in pregnancy complicated by GDM.

Our work pointed at increased GV in relation to the risk factors such as the FH of diabetes, the maternal age at conception, the pre-pregnancy BMI, and the maternal weight gain. GV once evaluated by 7-day CGM (%TIR, %TAR, %CV) was associated with various risk factors of GDM. Pregnant women with risk factors have higher probability of severe GV with its consequences on maternal and fetal health state.

The pre-pregnancy BMI and the weight gain are modifiable risk factors. Pre-pregnancy BMI is important especially when counseling women planning a pregnancy. Excessive weight gain avoidance in pregnancy may be an effective strategy for prevention of GDM as rapid weight gain in early pregnancy may result in an early increase in insulin resistance that leads to the “exhaustion” of the B cell [36].

## Figures and Tables

**Figure 1 ijerph-18-03402-f001:**
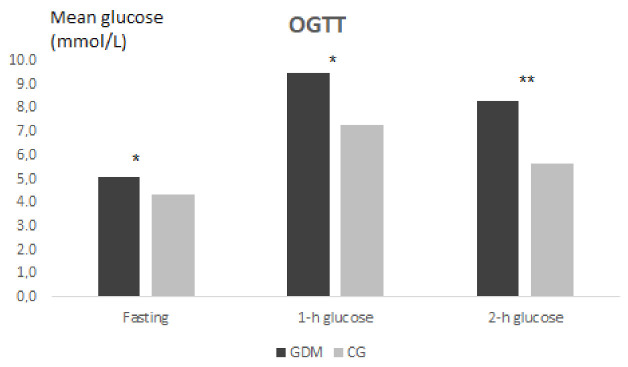
The comparison of the OGTT results in pregnant women with and without GDM (GDM/CG). The significance is set at *p* < 0.05. *p* value significance level: * 0.05, ** 0.01, *** 0.001.

**Figure 2 ijerph-18-03402-f002:**
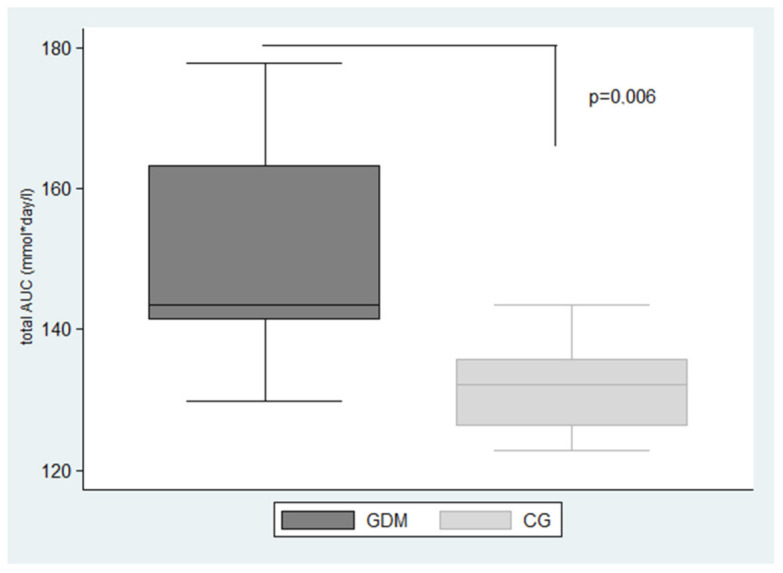
Box plot graph: the significant difference in glycaemic variability in GDM and CG compared by Total AUC (range 7.8–3.5 mmol/L). Data expressed as median (interquartile range): GDM, gestational diabetes mellitus; CG, control group; Total AUC, the area under the interstitial glucose concentration curve for the entire day. The significance is set at *p* < 0.05.

**Table 1 ijerph-18-03402-t001:** CGM-based TIR targets for GDM.

Diabetes Group	TIR	TAR	TBR
GDM	% of time in target range *: 3.5–7.8 mmol/L	<5% of time above target level 7.8 mmol/L	<4% of time below target level 3.5 mmol/L
			<1% of time below target level 3.0 mmol/L

CGM, continuous glucose monitoring; TIR, percentage of time in range; TAR, percentage of time above range; TBR, percentage of time below range. * There is lack of evidence on CGM percentage of time spent in range in GDM pregnancy, so this is not included in table. Modified from [14].

**Table 2 ijerph-18-03402-t002:** Baseline characteristics of pregnant women GDM/CG.

Baseline Characteristics (*n* = 20)	GDM (*n* = 11)	CG (*n* = 9)	*p*
Age (years)	30.63 ± 4.76	29.56 ± 2.56	0.44
Family history of diabetes (%)	90.90	33.30	0.009 **
Pre-pregnancy BMI (kg/m^2^)	25.4 ± 3.6	23.3 ± 3.9	0.31
Weight gain (kg)	9.8 ± 3.37	9.12 ± 3.34	0.52
FPG from OGTT (mmol/L)	5.04 ± 0.69	4.32 ± 0.44	0.03 *
1 h glucose (mmol/L)	9.46 ± 1.74	7.27 ± 0.55	0.002 **
2 h glucose (mmol/L)	8.25 ± 1.01	5.63 ± 0.56	0.0003 ***
C-peptide (ng/mL)	1.47 ± 0.5	1.67 ± 0.8	0.97
HbA1c NGSP ± SD NGSP (%)	5.52 ± 0.34	5.25 ± 0.2	0.09

Data expressed as mean ± standard deviation: GDM, gestational diabetes mellitus; CG, control group; BMI, body mass index; FPG, fasting plasma glucose; HbA1c, glycated haemoglobin A1c; OGTT, oral glucose tolerance test. The significance is set at *p* < 0.05. P value significance level: * 0.05, ** 0.01, *** 0.001.

**Table 3 ijerph-18-03402-t003:** Glycaemic variability in pregnant women GDM/CG.

CGM Indicators	GDM (*n* = 11)	CG (*n* = 9)	*p*
Gestational Age at CGM (week)	29.1 ± 2.34	29.33 ± 3.0	0.62
Mean glucose from sensor (mmol/L)	5.66 ± 0.58	5.46 ± 0.64	0.40
SD (mmol/L)	1.11 ± 0.36	0.81 ± 0.23	0.56
CV (%)	19.7 ± 6.43	15.04 ± 4.51	0.84
%TIR	94.63 ± 5.85	98.17 ± 1.72	0.08
%TAR	5.36 ± 5.85	1.67 ± 1.75	0.07
%TBR	0	0.11	0.27
J-index	0.046 ± 0.01	0.039 ± 0.01	0.14
Total AUC (mmol·day/L) **	149.41 ± 14.82	132.03 ± 6.93	0.0062 **
AUC_above7.8_ (mmol·day/L)	0.079 ± 0.11	0.015 ± 0.035	0.10
AUC_below3.5_ (mmol·day/L)	0	0	0

Data expressed as mean ± standard deviation unless otherwise stated. GDM, gestational diabetes mellitus; CG, control group; BMI, body mass index; CGM, continuous glucose monitoring; %CV, coefficient of variation; GA, gestational age; SD, standard deviation from sensor; %TAR, percentage of time above target range; %TIR, percentage of time in range; %TBR, percentage of time below target range; Total AUC, the area under the interstitial glucose concentration curve for the entire day AUC_above7.8_, the hyperglycaemic area under the interstitial glucose concentration curve above 7.8 mmol/L normalized for a 24-h period; AUC_below3.5_, the hypoglycaemic area under the interstitial glucose concentration curve below 3.5 mmol/L normalized for a 24-h period. The significance is set at *p* < 0.05. *p* value significance level: * 0.05, ** 0.01, *** 0.001.

**Table 4 ijerph-18-03402-t004:** Association between risk factors in pregnancy and GV.

Risk Factor	GV	*p*
Family history of diabetes	Total AUC (mmol·day/L) FPG from OGTT (mmol/L)	0.019 * 0.035 *
Pre-pregnancy BMI (kg/m^2^)	%TAR	0.054
Weight gain (kg)	%CV	0.030 *
Age (years)	%TIR %TAR AUC above 7.8 mmol/L (mmol·day/L)	0.003 ** 0.003 ** 0.042 * 0.005 **

BMI, body mass index; Total AUC, the area under the interstitial glucose concentration curve for the entire day; FPG, fasting plasma glucose; GV, glycaemic variability; OGTT, oral glucose tolerance test; %TIR, percentage of time in range; %TAR, percentage of time above range; %CV, coefficient of variation; AUC above 7.8 the hyperglycaemic area under the interstitial glucose concentration curve above 7.8 mmol/L normalized for a 24-h period. Non-parametric Spearman’s correlation was utilized to examine relationships between glycaemic control parameters and risk factors in all participants. The significance is set at *p* < 0.05. *p* value significance level: * 0.05, ** 0.01, *** 0.001.

## Data Availability

The supporting data in Excel are available on request. The source data are available in online Carelink software for diabetologist.

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
