# Peer review of "Glycaemic Variability and Risk Factors of Pregnant Women with and without Gestational Diabetes Mellitus Measured by Continuous Glucose Monitoring"

_ijerph, 2021, doi:10.3390/ijerph18073402_

Round 1
Reviewer 1 Report
Dr. Gáborová and colleagues present an original article on glycaemic variability and risk factors of pregnant women with and without gestational diabetes using continuous glucose monitoring. This manuscript offers new insights into an important topic and requires further attention. However, the flow, English, and grammar require significant review and would benefit from a native-English speaker reviewing the entire manuscript document. I have found some parts of the manuscript are difficult to read and understand. In addition, there are some comments that need to be addressed below.
Major and minor comments:
Line 38-41 needs to be re-worked. For example, CGM does not directly help “choose the most effective treatment method”. It can help with detection of glycemic excursions.
There should be p values in all of the tables when you have 2 or more comparisons. Table 1 and Table 2 should have additional columns with p values.
Please ensure that the acronyms being used are defined the first time and thereafter should only be the acronym. There are many situations where the acronym stops being used (e.g. time-in-range should be defined as TIR the first time, and TIR thereafter). On line 50, time-in-range should also include the acronym here and line 238 for example, should say TIR rather than time in range. Please be consistent with all of the acronyms throughout the manuscript.
If you define TBR on line 50, why is TIR and TAR not defined here as well? Please update.
Line 52 explanation of GDM should be re-worded.
SMBG stands for “self-monitoring of blood glucose” – please update on line 72.
There should be error around both the TIR and TAR in Table 2.
There is no mention of similar work published in Diabetes Technol and Therapeutics. Please review “Continuous Glucose Monitoring Versus Self-Monitoring of Blood Glucose to Assess Glycemia in Gestational Diabetes” (Zaharieva et al., Diabetes Technol Ther, 2019).
Please avoid the use of terms “good” and “bad” such as “bad glycaemic compensation” on Line 226. In addition, you make no mention of the demographics of the older pregnant women (Lines 224-227). How old were these women?
Author Response
Dear Reviewer 1,
I incorporated all comments and suggestions. Thank you very much for them, they helped a lot to improve our article special thank for the work Zaharieva et al.
Sincerely,
Gáborová Martina.
Dr. Gáborová and colleagues present an original article on glycaemic variability and risk factors of pregnant women with and without gestational diabetes using continuous glucose monitoring. This manuscript offers new insights into an important topic and requires further attention. However, the flow, English, and grammar require significant review and would benefit from a native-English speaker reviewing the entire manuscript document. I have found some parts of the manuscript are difficult to read and understand. In addition, there are some comments that need to be addressed below.
Major and minor comments:
Line 38-41 needs to be re-worked. For example, CGM does not directly help “choose the most effective treatment method”. It can help with detection of glycemic excursions.
The mentioned sentences were re-worked.
There should be p values in all of the tables when you have 2 or more comparisons. Table 1 and Table 2 should have additional columns with p values.
I added the p values in all the table.
Please ensure that the acronyms being used are defined the first time and thereafter should only be the acronym. There are many situations where the acronym stops being used (e.g. time-in-range should be defined as TIR the first time, and TIR thereafter). On line 50, time-in-range should also include the acronym here and line 238 for example, should say TIR rather than time in range. Please be consistent with all of the acronyms throughout the manuscript.
I made the acronyms consistent throughout the manuscript.
If you define TBR on line 50, why is TIR and TAR not defined here as well? Please update.
I have already defined them on one place.
Line 52 explanation of GDM should be re-worded.
I placed there new definition.
SMBG stands for “self-monitoring of blood glucose” – please update on line 72.
I did.
There should be error around both the TIR and TAR in Table 2.
I corrected it.
There is no mention of similar work published in Diabetes Technol and Therapeutics. Please review “Continuous Glucose Monitoring Versus Self-Monitoring of Blood Glucose to Assess Glycemia in Gestational Diabetes” (Zaharieva et al., Diabetes Technol Ther, 2019).
Thank you for this nice work, I appreciate it a lot.
Please avoid the use of terms “good” and “bad” such as “bad glycaemic compensation” on Line 226. In addition, you make no mention of the demographics of the older pregnant women (Lines 224-227). How old were these women?
These women were: mean age ±SD (3.6 ±2.19).
Reviewer 2 Report
Gaborova and colleagues compared in their study the glycaemic variability and risk factors of diabetes in pregnant women with and without gestational diabetes mellitus.
Introduction
Please shift the first sentence of the materials and methods section to the last paragraph of the introdcution section. The aim of the study should be described at the end of the introduction.
Materials and Methods
At what time the study was carried out? Please add at least the years.
Was the study registered before study start in a clinical trials database e.g. clinicaltrials.gov? Please add the registration number. If it has been not registered please explain why not.
Please add the ethics number of the approval by the Ethics committee of Kosice region and Pavol Jozef Safarik University.
Line 88: 20 pregnant women .....
Line 124: A total of 23 pregnant women.... the number of patients should match, please check.
Line 90: The anamnestic and biochemical parameters were taken. Please replace "anamnestic" with "demographic".
Line 94: Please add the citation for the criteria of Slovak Diabetes Society 2018 for gestational diabetes mellitus.
Line 96: please shit the results in the results part and please make sure that the data match. there are some values which do not match between text and table. (e.g. HbA1c 5,52 in text vs. 5,50 in table 1 in the GDM-group; HbA1c 5,25 in text vs. 5,20 in table 1).
Line 97: please include the unit "years" after age
Line 95: The results were written as mean+- standad deviation.....in table 1 data are axepressed as mean+- standard error! Please clarify what is correct and why you are using the standard error of mean in the table?
Please explain how the Area under the curve (AUC) was calculated?
Table 1: please add a column with the p-values for group comparison for each listed parameter in the table.
Line 112 and Table 1 Heading: baseline characteristics, please add a "s" in the heading as well as in the heading of the table.
RESULTS
Line 129: please change table 2 in table 1.
Please add a figure (curve) of oral glucose tolerance tests comparing both groups, so it would be easier to compare the both groups. I
Line 153: ...familiy history of diabetes even in one pregnant women. Please change to "one pregnant woman", further please check and compare to the information in the table 1. There are 33,30 % with family history of diabetes in the control group - these should be three women, please check!
Why the cohort size is 20? Is there a sample size calculation? Please explain and add to the methods part!
Author Response
Dear Reviewer 2,
I considered all comments and suggestions. Thank you very much for your detailed rewiev. It helped a lot to improve our article.
Sincerely,
Gáborová Martina.
Gaborova and colleagues compared in their study the glycaemic variability and risk factors of diabetes in pregnant women with and without gestational diabetes mellitus.
Introduction
Please shift the first sentence of the materials and methods section to the last paragraph of the introdcution section. The aim of the study should be described at the end of the introduction.
Thank you, I did.
Materials and Methods
At what time the study was carried out? Please add at least the years.
Yes, very good comment the study was conducted in 2019-2020.
Was the study registered before study start in a clinical trials database e.g. clinicaltrials.gov? Please add the registration number. If it has been not registered please explain why not.
Please add the ethics number of the approval by the Ethics committee of Kosice region and Pavol Jozef Safarik University.
The study was not registered in a clinical trials database.
The study was approved by the dean of Pavol Jozef Šafárik University Medical faculty in Košice and Ethics Commission of the Košice Self-Governing Region, protocol number: 8N/ 2019.
Line 88: 20 pregnant women .....
Line 124: A total of 23 pregnant women.... the number of patients should match, please check.
All subjects (23 pregnant women) provided written informed consent before participating in the study.
Three out of 23 participants had inaccurate CGM results due to the sensor's low intrinsic signal and they were excluded from analyses.
Line 90: The anamnestic and biochemical parameters were taken. Please replace "anamnestic" with "demographic".
Thank you. I did.
Line 94: Please add the citation for the criteria of Slovak Diabetes Society 2018 for gestational diabetes mellitus.
Thank you. I did.
Line 96: please shit the results in the results part and please make sure that the data match. there are some values which do not match between text and table. (e.g. HbA1c 5,52 in text vs. 5,50 in table 1 in the GDM-group; HbA1c 5,25 in text vs. 5,20 in table 1).
I am sorry for this was mistake. I re-worked it.
Line 97: please include the unit "years" after age
Yes, thank you, I did.
Line 95: The results were written as mean+- standad deviation.....in table 1 data are axepressed as mean+- standard error! Please clarify what is correct and why you are using the standard error of mean in the table?
This was also mistake. I re-worked it. I meant standard deviation.
Please explain how the Area under the curve (AUC) was calculated?
Thank you, I placed the equation for AUC into the methods. I added equation for coefficient of variation, too.
Table 1: please add a column with the p-values for group comparison for each listed parameter in the table.
I added the p values in all tables.
Line 112 and Table 1 Heading: baseline characteristics, please add a "s" in the heading as well as in the heading of the table.
Thank you, I did.
RESULTS
Line 129: please change table 2 in table 1.
I re-worked it.
Please add a figure (curve) of oral glucose tolerance tests comparing both groups, so it would be easier to compare the both groups. I
I added the figure of oral glucose tolerance test to compare the results of both groups.
Line 153: ...familiy history of diabetes even in one pregnant women. Please change to "one pregnant woman", further please check and compare to the information in the table 1. There are 33,30 % with family history of diabetes in the control group - these should be three women, please check!
It was serious mistake. I re-worked it.
Why the cohort size is 20? Is there a sample size calculation? Please explain and add to the methods part!
The cohort was small because we had to exclude women which failed their SMBG, women with low intrinsic sensor signal or different types of diabetes. The observation was difficult as they had to agree with 7-day CGM and detail SMBG.
Reviewer 3 Report
The authors address an important issue namely how CGM can assess glycaemic control and specifically glycaemic variability in GDM as well as some of the underlying risk factors and determinants.
This is a small study however there is quite a lot of data and the analysis goes beyond the customary Time In Range, Time Above Range and Time Below Range so the rationale for these additional parameters eg AUC and what they show and how useful this is if at all needs to be discussed as that is part of how this study is different from others reported. HbA1c is a measure of average glucose control rather than glycaemic variability so that point should be made. the J index needs to be explained briefly when first mentioned. The abbreviations TIR, TAR and TBR need to all be shown in the second paragraph of the introduction.
The introduction and discussion should include reference to these recent papers:
Continuous Glucose Monitoring in Pregnancy: Importance of Analyzing Temporal Profiles to understand Clinical Outcomes E M Scott et al Diabetes Care 2020;43:1178-1184.
Application and Utility of Continuous Glucose Monitoring in Pregnancy: A Systematic Review Qi Yu et al Frontiers in Endocrinology doi: 10.3389/fendo.2019.00697
The last one or two sentences of the introduction need to state clearly the objectives of the study.
The text needs to refer more clearly to the tables and figures. Table 1 needs to be mentioned in the second paragraph of materials and methods and deleted from the last sentence of the fourth paragraph of materials and methods.
Table 2 should be at the end of the sentence "The CGM indicators of glucose variability... are presented in table 3" - it should be "in table 2"
You need to refer to Table 3 in the text - at the end of the sentence "Other parameter of glycaemic variability from CGM such as...."
You need a further table 4 to show the correlations you describe in the text
Most of the discussion is acceptable. You need to make clear what is new in your study versus what is already known.
A major revision of the English will result in a much better paper.
Author Response
Dear Reviewer 3,
I incorporated all comments and suggestions. Thank you very much for them, they helped a lot. Special thanks for the work Scott et al and Qi Yu et al.
Sincerely,
Gáborová Martina.
The authors address an important issue namely how CGM can assess glycaemic control and specifically glycaemic variability in GDM as well as some of the underlying risk factors and determinants.
Thank you for this comment.
This is a small study however there is quite a lot of data and the analysis goes beyond the customary Time In Range, Time Above Range and Time Below Range so the rationale for these additional parameters eg AUC and what they show and how useful this is if at all needs to be discussed as that is part of how this study is different from others reported. HbA1c is a measure of average glucose control rather than glycaemic variability so that point should be made. the J index needs to be explained briefly when first mentioned. The abbreviations TIR, TAR and TBR need to all be shown in the second paragraph of the introduction.
Thank you, I have already made the acronyms consistent throughout the manuscript.
The introduction and discussion should include reference to these recent papers:
Continuous Glucose Monitoring in Pregnancy: Importance of Analyzing Temporal Profiles to understand Clinical Outcomes E M Scott et al Diabetes Care 2020;43:1178-1184.
Application and Utility of Continuous Glucose Monitoring in Pregnancy: A Systematic Review Qi Yu et al Frontiers in Endocrinology doi: 10.3389/fendo.2019.00697
Thank you, I incorporated these works.
The last one or two sentences of the introduction need to state clearly the objectives of the study.
Thank you, I did.
The text needs to refer more clearly to the tables and figures. Table 1 needs to be mentioned in the second paragraph of materials and methods and deleted from the last sentence of the fourth paragraph of materials and methods.
Table 2 should be at the end of the sentence "The CGM indicators of glucose variability... are presented in table 3" - it should be "in table 2"
You need to refer to Table 3 in the text - at the end of the sentence "Other parameter of glycaemic variability from CGM such as...."
I did. I am sorry, there were some mistakes. I have already put the tables in order.
You need a further table 4 to show the correlations you describe in the text
I did. Table 4. Association between risk factors in pregnancy and parameters of glycaemic control.
Most of the discussion is acceptable. You need to make clear what is new in your study versus what is already known.
Thank you. I did.
A major revision of the English will result in a much better paper.
Yes, I did.
Round 2
Reviewer 1 Report
Thank you Dr. Gáborová and colleagues for the revisions on your manuscript. This manuscript offers valuable information to the field. However, the flow, English, and grammar still require significant review and it does not appear that a native-English speaker has reviewed the entire manuscript document. As you can see in the comments below, there are still a significant number of grammatical and spelling errors noted in the manuscript. Once addressed, the manuscript will read much easier. Please address the remaining comments.
Comments:
Line 14: Please define GDM in the abstract.
Line 23: Change weigh to weight.
Line 26: Please change to “…a family history” and again weight not weigh.
Line 27: Please re-word the last sentence. You should not “assume”. It should either shown in the data or not.
Line 32-33: You use “used” twice in the same sentence so please remove one and may need to re-word this sentence.
The first paragraph (specifically Lines 32-38) describing CGM could be strengthened. The language is not very clear and could benefit with review for the flow.
Line 44: Change to “…by the beta cells of the pancreas”
Line 45: Remove “the” in “the normal glucose…”
Line 45: Please change the word “riskiest” – highest risk??
Line 53-54: An obesity? Please fix this.
Line 63: I wouldn’t necessarily say a more accurate picture of glycaemic control because this sounds like CGM is more accurate than SMBG and this is not the comparison. I would re-word this sentence. It definitely tells us more information by capturing more frequent glucose values.
Line 67: Change to say something like “possible bleeding from sensor insertion”
Line 67: Please change CCM to CGM and check all acronyms throughout the paper.
Line 69-73: I think this sentence needs to be updated. You should be citing the official CGM consensus guidelines and targets, particularly in pregnancy: Clinical Targets for Continuous Glucose Monitoring Data Interpretation: Recommendations from the International Consensus on Time in Range – Battelino et al. 2019 Diabetes Care.
Line 77: Glycemic should be spelled glycaemic and kept consistent throughout based on journal submission guidelines. Please check through the entire manuscript.
Line 79: Update twenty-three to 23. Typically, numbers >10 can be used in numerical terms and <10 should be spelled out in words.
Line 80: Update to “after an oral glucose tolerance test”
Line 82: dean should be spelled Dean and typically the Dean does not approve research studies, but the ethics committee does (as mentioned). Please update accordingly unless this is different at your institute.
Line 86: Change volunteers to participants
Line 87: Why glucose oral test? You defined OGTT above, please use consistent terms.
Line 88: Please fix the grammar and punctuation errors… “7.8 mmol/l.) if not they were…” This should be 2 sentences.
Line 90: What bleeding disorders? May want to use specific medical terminology here.
Line 92: Update “After the OGTT,…”
Line 93: Continuous glucose monitoring is already defined. Please be more careful throughout the manuscript for acronyms already defined!
Line 93: Typically, after the product such as iPro2 Professional, you should state the brand and headquarters in brackets.
The investigated CGM parameters: This entire section you sometimes use commas and sometimes decimals for glucose concentrations. This needs to be consistent. The same applies for Table 1.
Table 1 is quite unclear because you don’t write mmol/l in each column, so it is difficult to know what we are looking at. You mention in brackets in the first column (mmol/l), but the glucose is NOT in brackets in the remaining columns.
Line 184: Update “In the GDM group…” and 6 should be “six” and 4 should be “four” etc.
Table 3: Please spell out Gestational Age and do not use GA – this is not a very common acronym and is confusing. If you use acronyms, they should be spelled out in order in the caption. Also, AUCabove7.8 is unclear in the caption – please add units.
Something is not adding up in Table 3. You report TIR and TAR for both groups, but no TBR. You mention no one had TBR? If this is the case, why does your TIR and TAR not add up to 100 in either group? Please check your numbers and add TBR because it appears there is ~2% missing in each group.
For all tables, significance should be denoted either by bold or an asterisks (*) or some other method. In addition, you should mention in all table captions that significance was set at P<0.05 so readers do not have to scroll through the paper to see what you set significance at.
Table 4 is confusing to follow. You show P values and no data which is not common. Are these correlations? Please consider removing or listing in the text rather than in a table.
Line 266: Please fix “60. minute” and remove in? This sentence is currently unclear.
Line 274: “In the GDM group”
Line 276: HbA1c not HbA1…
Line 287: Missing a period.
Line 288: This is an incomplete sentence “We also that….”
Line 292: What does “It has a low interference with e.g. anaemia or age” mean? Please make this sentence clearer or remove.
Line 294: You already defined SMBG. And SMBG does not TREAT diabetes. Be careful with your language. It is simply a management tool for measuring blood glucose.
Your conclusion does not mention CGM at all and the primary focus of your paper is the important findings related to CGM. Please add more detail and add a limitations section within the discussion.
Author Response
Dear Reviewer 1,
I re-worked my article according to your comments and suggestions.
Thank you.
Sincerely,
Gáborová Martina.
Thank you Dr. Gáborová and colleagues for the revisions on your manuscript. This manuscript offers valuable information to the field. However, the flow, English, and grammar still require significant review and it does not appear that a native-English speaker has reviewed the entire manuscript document. As you can see in the comments below, there are still a significant number of grammatical and spelling errors noted in the manuscript. Once addressed, the manuscript will read much easier. Please address the remaining comments.
Comments:
Line 14: Please define GDM in the abstract.
Revised as requested.
Line 23: Change weigh to weight.
Done.
Line 26: Please change to “…a family history” and again weight not weigh.
Thank you. Revised as requested.
Line 27: Please re-word the last sentence. You should not “assume”. It should either shown in the data or not.
Thank you. I re-worked that sentence.
Line 32-33: You use “used” twice in the same sentence so please remove one and may need to re-word this sentence.
I have already re-worded the mentioned sentence.
The first paragraph (specifically Lines 32-38) describing CGM could be strengthened. The language is not very clear and could benefit with review for the flow.
I completely re-worded it. Thank you for this comment.
Line 44: Change to “…by the beta cells of the pancreas”
Revised as requested.
Line 45: Remove “the” in “the normal glucose…”
Revised as requested.
Line 45: Please change the word “riskiest” – highest risk??
Thank you very much. I have already re-worded that.
Line 53-54: An obesity? Please fix this.
Done.
Line 63: I wouldn’t necessarily say a more accurate picture of glycaemic control because this sounds like CGM is more accurate than SMBG and this is not the comparison. I would re-word this sentence. It definitely tells us more information by capturing more frequent glucose values.
Very good comment. Thank you. I have re-worded that.
Line 67: Change to say something like “possible bleeding from sensor insertion”
I deleted that sentence.
Line 67: Please change CCM to CGM and check all acronyms throughout the paper.
Of course, thank you, I am sorry for such mistake.
Line 69-73: I think this sentence needs to be updated. You should be citing the official CGM consensus guidelines and targets, particularly in pregnancy: Clinical Targets for Continuous Glucose Monitoring Data Interpretation: Recommendations from the International Consensus on Time in Range – Battelino et al. 2019 Diabetes Care.
Thank you. I implemented these official guidelines into my results and tables.
Line 77: Glycemic should be spelled glycaemic and kept consistent throughout based on journal submission guidelines. Please check through the entire manuscript.
Done.
Line 79: Update twenty-three to 23. Typically, numbers >10 can be used in numerical terms and <10 should be spelled out in words.
Thank you. Done.
Line 80: Update to “after an oral glucose tolerance test”
Thank you. Done.
Line 82: dean should be spelled Dean and typically the Dean does not approve research studies, but the ethics committee does (as mentioned). Please update accordingly unless this is different at your institute.
In reality the study was approved by the Ethics Committee of the Košice Self-Governing Region, protocol number: 8N/ 2019. The Dean of Medical Faculty just gave a recommendation.
Line 86: Change volunteers to participants
Thank you. Done.
Line 87: Why glucose oral test? You defined OGTT above, please use consistent terms.
Thank you very much. Revised as requested.
Line 88: Please fix the grammar and punctuation errors… “7.8 mmol/l.) if not they were…” This should be 2 sentences.
Thank you very much. I corrected it.
Line 90: What bleeding disorders? May want to use specific medical terminology here.
I deleted this sentence because bleeding disorders are quite rare in pregnant women.
Line 92: Update “After the OGTT,…”
Thank you very much. I fixed that.
Line 93: Continuous glucose monitoring is already defined. Please be more careful throughout the manuscript for acronyms already defined!
Yes, that is true. Thank you very much I did´t notice that.
Line 93: Typically, after the product such as iPro2 Professional, you should state the brand and headquarters in brackets.
Thank you. Yes, I added it.
The investigated CGM parameters: This entire section you sometimes use commas and sometimes decimals for glucose concentrations. This needs to be consistent. The same applies for Table 1.
Ok. Thank you. I have made it consistent. I completely re-worked Table 1.
Table 1 is quite unclear because you don’t write mmol/l in each column, so it is difficult to know what we are looking at. You mention in brackets in the first column (mmol/l), but the glucose is NOT in brackets in the remaining columns.
I completely re-worked Table 1.
Line 184: Update “In the GDM group…” and 6 should be “six” and 4 should be “four” etc.
Thank you very much. I fixed that.
Table 3: Please spell out Gestational Age and do not use GA – this is not a very common acronym and is confusing. If you use acronyms, they should be spelled out in order in the caption. Also, AUCabove7.8 is unclear in the caption – please add units.
Thank you very much. I corrected it.
Something is not adding up in Table 3. You report TIR and TAR for both groups, but no TBR. You mention no one had TBR? If this is the case, why does your TIR and TAR not add up to 100 in either group? Please check your numbers and add TBR because it appears there is ~2% missing in each group.
In our participants TBR was present in only one woman in CG so I didn´t show these results. TBR was falsely positive in GDM and CG, too (by checking ISIG in concrete hypoglycaemias we found they are false) The result was that TIR was not corrected, I am very sorry I re-worked TIR and TBR completely according to the current results.
For all tables, significance should be denoted either by bold or an asterisks (*) or some other method. In addition, you should mention in all table captions that significance was set at P<0.05 so readers do not have to scroll through the paper to see what you set significance at.
Thank you very much for this comment. I revised that.
Table 4 is confusing to follow. You show P values and no data which is not common. Are these correlations? Please consider removing or listing in the text rather than in a table.
These are correlations. Reviewer 3 wanted me to put these correlations into the table for a better clarity.
Line 266: Please fix “60. minute” and remove in? This sentence is currently unclear.
I removed it.
Line 274: “In the GDM group”
Done.
Line 276: HbA1c not HbA1…
Done. Thank you.
Line 287: Missing a period.
Done.
Line 288: This is an incomplete sentence “We also that….”
I re-worded it.
Line 292: What does “It has a low interference with e.g. anaemia or age” mean? Please make this sentence clearer or remove.
I removed it. Thank you.
Line 294: You already defined SMBG. And SMBG does not TREAT diabetes. Be careful with your language. It is simply a management tool for measuring blood glucose.
I corrected it.
Your conclusion does not mention CGM at all and the primary focus of your paper is the important findings related to CGM. Please add more detail and add a limitations section within the discussion.
Thank you for this comment. I added more concrete conclusions to the conclusion part. I added strengths and limitations of the study to the discussion, too.
Reviewer 2 Report
Thank you very much for the answer to my suggestions and the editing in the manuscript.
Author Response
Dear Reviewer 2,
thank you very much for your quick response. I appreciate it a lot.
Sincerely,
Gáborová Martina.
Reviewer 3 Report
The manuscript is much improved
Abstract
line 10-11 suggest were not significantly different
Line 11 should read weight gain
Author Response
Dear Reviewer 3,
I re-worked my article according to your comments and suggestions.
Thank you.
Sincerely,
Gáborová Martina.
The manuscript is much improved
Thank you very much.
Abstract
line 10-11 suggest were not significantly different
I revised that.
Line 11 should read weight gain
Done. Thank you.